# The effect of COVID-19 vaccination on the menstrual pattern and mental health of the medical students: A mixed-methods study from a low and middle-income country

Rabia Kareem[1], Mifrah Rauf Sethi[2], Sumaira Inayat[3], Muhammad Irfan[2]*

1 Department of Obstetrics and Gynecology, Peshawar Medical College, Riphah International University, Islamabad, Pakistan, 2 Department of Mental Health, Psychiatry and Behavioral Sciences, Peshawar Medical College, Riphah International University, Islamabad, Pakistan, 3 Department of Obstetrics and Gynecology, Northwest General Hospital and Research Center, Peshawar, Pakistan

* mirfan78@yahoo.com

## Abstract

**Data Availability Statement:** All relevant data are within the paper.

### Objective

To assess the effect of COVID-19 vaccination on menstrual patterns and mental health of medical students and to explore the students' perspective regarding this effect.

### Materials and methods

This mixed-method study was conducted on the medical and dental students of the private and public sector institutions of Peshawar from September 2021 to March 2022. A Menstrual symptom questionnaire (MSQ) and hospital anxiety and depression scale (HADS) were used. This was followed by qualitative interviews with the students who faced problems in their menstruation after the COVID-19 vaccination.

### Results

A total of 953 students were included, with a mean age of 20.67±1. 56 years. More than half (n = 512, 53.7%) experienced menstrual cycle abnormalities post-vaccination. The majority having disturbances in their menstrual cycle had significantly higher levels of anxiety (p = 0.000). Results on the menstrual symptom questionnaire, anxiety, and depression subtype of HADS showed a negative and statistically significant relationship with changes after COVID-19 vaccination (p<0.05). In the qualitative interviews, 10 (58.8%) students each had problems with frequency and flow, followed by 7 (41.2%) students, who had dysmenorrhea. Seven (41.2%) consulted a gynecologist for management. The majority (n = 14, 82.4%) stated that these issues had an adverse impact on their mental health and almost half (n = 8, 47.1%) suggested consulting a gynecologist while facing such situations.

**Funding:** The author(s) received no funding for this work.

**Competing interests:** The authors have declared that no competing interests exist.

## Conclusion

This study showed the impact of the COVID-19 vaccine on women's menstrual patterns and subsequent mental health status. Although the majority of the students experienced menstrual cycle abnormalities and subsequent mental health adversities post COVID-19 vaccination but these were temporary and self-limiting and were attributed to the psychological impact of the vaccination. Therefore, it is imperative to alert health care professionals about possible side effects and prior counseling is expected to play an important role in this context.

## Introduction

Over the past two years, the COVID-19 pandemic has taken the world by storm, affecting millions of people directly and billions indirectly, making it one of the deadliest in history [1, 2]. In the face of these global challenges, scientists worldwide joined hands to develop COVID-19 vaccines in less than a year from when the virus was first identified, and since then it has shown great progress in managing the pandemic [3, 4]. Currently, over nine billion doses of COVID-19 vaccines have been administered. Many studies reported a variety of vaccine-related side effects ranging from mild symptoms like fever and injection site pain to more serious and long-lasting complications such as cardiovascular complications including myocardial infarction and stroke [1, 5]. Recently, there have also been reports on menstrual irregularities post-COVID-19 vaccination including changes in the cycle length, duration, flow, dysmenorrhea, and abnormal uterine bleeding [6]. Although a statement issued by the pharmacovigilance assessment committee of the European Medical Agency said that no association between COVID-19 vaccination and menstruation has been found but the Medicines and Health Care Products Regulatory Agency in the United Kingdom reported a good number of cases of menstrual problems [1, 7].

There has been an increasing public concern and fears about the effects of COVID-19 vaccination on menstruation. Still, currently, there is limited data for investigating this relationship which may also include its impact on mental health [4, 8]. The pandemic itself has affected the reproductive health of the women which is associated with a significant increase in suffering from mental health symptoms [9].

Since the menstrual cycle, is an indicator of the reproductive health of a woman, it is considered as a proxy measure of health and fertility, and women with menstrual irregularities are more prone to metabolic disorders [10, 11]. Therefore, the lack of evidence about the impact of COVID-19 vaccination on the menstruation, in specifically this part of the world limits our ability to sufficiently address the possible concerns of those receiving vaccination, which may be a reason of vaccine hesitation among women of reproductive age group and a cause of fear about its impact on their fertility [12]. Based on these facts, we decided to investigate the effects of COVID-19 vaccination on the menstrual pattern and mental health of the medical and dental students of Peshawar and to explore the students' perspectives regarding this effect.

To the best of our knowledge, this study will be the first of its kind to evaluate the impact of the vaccine not only on physical but on mental health as well. Furthermore, it will guide the research community to explore the phenomenon further.

## Materials and methods

This convergent sequential-explanatory mixed-method study was conducted on medical and dental students, between the age of 18 to 25 years, at different private and public sector

institutions in Peshawar, Pakistan from September 2021 to March 2022. All female students with a previously normal menstrual cycle (before the COVID-19 vaccination) were included while students with mental health issues were excluded from the study. It was conducted in two phases. During Phase 1(Quantitative part) pre-designed questionnaires were distributed among the medical and dental students. In phase 2 (Qualitative part) interviews were conducted with those students who had menstrual irregularities.

## Phase-1

A cross-sectional survey was conducted including all female students who received COVID-19 vaccination. A purposive sampling technique was used. Ethical Approval was taken from Prime Foundation (PRIME/IRB/2021-330), Pakistan, and permission was obtained from the concerned authorities of all the institutions visited for data collection. All the students who participated in the study were informed about the objectives of the study and written informed consent was obtained. Demographic data was collected on a structured proforma before collecting the data, followed by applying the following scales.

**Instruments.**   *Menstrual Symptom Questionnaire (MSQ).* Menstrual symptoms were measured using the MSQ, which is a 24-item self-report measure that assesses menstrual pain and symptoms. The score on each item ranges from 1 (never) to 5 (always) with a higher composite score indicating more symptoms [13].

*Hospital Anxiety and Depression Scale (HADS).* The HADS aim to measure symptoms of anxiety and depression and consists of 14 items, seven items for the anxiety subscale (HADS Anxiety) and seven for the depression subscale (HADS Depression). Each item is scored on a response-scale with four alternatives ranging between 0 and 3. After adjusting for six items that are reverse scored, all responses are summed to obtain the two subscales. Recommended cut-off scores according to Zigmond & Snaith are 8–10 for doubtful cases and ≥11 for definite cases. An optimal balance between sensitivity and specificity was found using a cut-off score of 8 or above for both HADS Anxiety and HADS Depression [14].

## Phase 2

This was followed by qualitative interviews with 17 medical and dental students. Students with menstrual irregularities who approached the primary investigator and the co-investigator were interviewed to explore their perceptions of the effects of COVID-19 vaccination on menstruation. The interview guide was developed by following the protocol for questionnaire development and was validated by experts.

The analysis of the data was carried out using SPSS v.25. Basic variables were analyzed using descriptive statistics for finding frequencies and percentages, means, and standard deviation. The internal consistency of the scales was measured through Cronbach's Alpha reliability and alpha equal to or greater than 0.70 was considered satisfactory. A Chi-square test was used to know the different types of COVID-19 vaccine with anxiety and depressive subtypes of HADS. An independent sample t-test was used to find out the mean differences based on subtypes of HADS (Anxiety & Depression) with the menstrual symptoms questionnaire. In addition, the Pearson correlation test was used to check the relationship between all scales and different types of vaccines. The results of all the tests of significance were considered significant at $p < 0.05$ level.

For the qualitative part, we used thematic analysis using Virginia Braun and Victoria Clarke six steps approach to generate themes from the qualitative data. The six steps included familiarization with the data, generating initial codes, searching for themes followed by reviewing, defining, and naming the themes, and finally producing the report [15].

## Results

A total of 1025 were asked to participate in the study. The response rate was 93% (n = 953). The mean age of the sample was 20.67 ± 1.56 years with the age range of 18–24 years. The Cronbach Alpha Reliability of the Menstrual Symptom Questionnaire was 0.897 and the Hospital Anxiety and Depression scale were 0.755. Majority of the students were from pre-clinical years (n = 678, 71.1%), and from Private sector institution (n = 500, 52.5%) respectively. Regarding any history of COVID-19 illness in the past, most of the students were not affected by it (n = 633, 66.4%). Most of the students received complete doses of COVID-19 vaccination (n = 857, 89.9%). Most of the students got Sinovac Vaccination (n = 450, 47.2%), followed by Sinopharm (n = 306, 32.1%) respectively. According to the responses on the Hospital Anxiety and Depression Scale (HADS), the majority had anxiety (n = 610, 64%) while only 101 (10.6%) had depression. Out of 610 students, that showed symptoms of anxiety, 291 received Sinovac vaccination, 198 received Sinopharm, 50 received Pfizer, 19 received Moderna, 12 received Cansino, 7 received AstraZeneca, and 33 of the students did not know the brand name of the vaccine received. Out of 101 students, that showed symptoms of depression, 38 received Sinovac vaccination, 33 received Sinopharm, 9 received Pfizer, 8 received Moderna, 3 received AstraZeneca, 1 received Cansino, and 9 of the students did not know the brand name of the vaccine received. Similarly, on MSQ, more than half of the students had disturbances in their menstrual cycle (n = 512, 53.7%). Out of these 512 students, 257 received Sinovac, 156 received Sinopharm, 43 received Pfizer, 16 received Moderna, 7 received Cansino, only 4 received AstraZeneca and 29 of the students did not know the brand name of the vaccine respectively. The Complete details are given in Table 1.

**Table 1. Basic demographic details of the study (n = 953).**

| S. No | Variables | | n (%) |
|---|---|---|---|
| 1 | Institution | Private | 500 (52.5%) |
| | | Public | 453 (47.5%) |
| 2 | Year | Pre-Clinical Years | 678 (71.1%) |
| | | Clinical Years | 275 (28.9%) |
| 3 | History of COVID illness | Yes | 320 (33.6%) |
| | | No History | 633 (66.4%) |
| 4 | Doses of Vaccination | Single Dose | 96 (10.1%) |
| | | Both Doses | 857 (89.9%) |
| 5 | Type Of Vaccination | AstraZeneca | 12 (1.3%) |
| | | Cansino | 16 (1.7%) |
| | | Moderna | 28 (2.9%) |
| | | Pfizer | 78 (8.2%) |
| | | Sinopharm | 306 (32.1%) |
| | | Sinovac | 450 (47.2%) |
| | | Don't Know | 63 (6.6%) |
| 6 | Anxiety Subtype of Hospital Anxiety and Depression Scale | Normal | 121 (12.7%) |
| | | Borderline Abnormal | 222 (23.3%) |
| | | Abnormal | 610 (64%) |
| 7 | Depression Subtype of Hospital Anxiety and Depression Scale | Normal | 475 (49.8%) |
| | | Borderline Abnormal | 377 (39.6%) |
| | | Abnormal | 101 (10.6%) |
| 8 | Menstrual Symptom Questionnaire | No Disturbance | 441 (46.3%) |
| | | Disturbance | 512 (53.7%) |

**Table 2. Mean difference and t-value of subtypes of Hospital Anxiety and Depression Scale on menstrual symptom questionnaire (n = 512).**

| | Menstrual Symptom Questionnaire | t-value | Mean Difference | 95% Confidence Interval | | p-Value |
|---|---|---|---|---|---|---|
| | M±SD | | | Lower | Upper | |
| Anxiety subtype of HADS | 12.29±2.89 | **-8.27**** | -1.61 | -1.99 | -1.22 | **.000**** |
| Depression subtype of HADS | 7.86±2.46 | **-4.59**** | -.732 | -1.04 | -.419 | **.000**** |

Note

** p < 0.05.

The results based on the Chi-square test showed no significant difference between students of clinical and pre-clinical years as far as menstrual disturbances were concerned (p = 0.184). However, the menstrual cycle was significantly more disturbed in private sector students as compared to public sector students (p = 0.017).

Out of 512 participants that showed menstrual disturbance, 382 (74.6%) had anxiety and 64 (12.5%) had depression, when HADS was applied, excluding borderline abnormal cases (91 (17.8%) and 223 (43.6%) cases, respectively). The results of the independent sample t-test on the anxiety and depression subtype of HADS with menstrual symptom questionnaire showed statistically significant results (p = 0.000). The majority of the females, who have disturbances in their menstrual cycles have a significantly higher level of anxiety (p = 0.000) but fall within the normal range on the depression subtype of HADS (p = 0.000) respectively. Detailed results are given in Table 2.

Pearson coefficient correlation test was applied to see the relationship between subtypes of HADS, menstrual cycle with types of COVID-19 vaccinations. The results generally revealed a negative and non-significant relationship between subtypes of HADS and menstrual cycle with the types of vaccination (p>0.05), whereas showed a positive and statistically significant relationship between anxiety and depression subtypes with menstrual cycle disturbances (p<0.05) respectively. Complete details are given in Table 3.

## Qualitative analysis

The following themes emerged during the interviews with the students (n = 17) who came to the primary investigator and the corresponding author, after the survey, with the menstrual problems they faced after the COVID-19 vaccination.

**Theme 1: Problems with menstrual patterns after covid-19 vaccination.** The majority of the students suffered from changes in the frequency and flow of the menstrual cycle (n = 10,

**Table 3. Pearson correlation between subtypes of Hospital Anxiety and Depression Scale and menstrual symptom questionnaire with types of COVID-19 vaccination (n = 953).**

| S. No | VARIABLES | I | II | III | IV |
|---|---|---|---|---|---|
| I | Types of Vaccines | 1 | | | |
| II | Anxiety subtype of Hospital Anxiety and Depression Scale | -.046 | 1 | | |
| | | (.155) | | | |
| III | Depression subtype of Hospital Anxiety and Depression Scale | -.015 | .293** | 1 | |
| | | (.647) | (.000) | | |
| IV | Menstrual Symptom Questionnaire | .032 | .329** | .138** | 1 |
| | | (.330) | (.000) | (.000) | |

Note

** p < 0.05.

58.8% each), followed by an increase in dysmenorrhea (n = 7, 41.2%) and a change in the length of bleeding days (n = 6, 35.3%). Few students reported an increase in the severity of pre-menstrual syndrome (n = 5, 29.4%), and 4 (23.5%) observed changes in the regularity of the cycles. Only 2 (11.8%) experienced intermenstrual bleeding. The problems and disturbances varied in nature which is apparent from the verbatim below.

*"Initially, after the first dose, my cycles got delayed and after I had my periods, I used to experience intermenstrual spotting. This also made me anemic and slowly my skin deteriorated with it. The flow became very heavy. I also experienced severe dysmenorrhea for which I even had to miss college at times."* (S4)

*"After vaccination, the dysmenorrhea increased and it was unbearable for me. I used to feel like pulling my hair. The flow was very heavy and was filled with clots and I experienced numbness in the peripheries."* (S11)

*"Ever since I have been vaccinated. . . I had fatigue with loss of appetite and a lot of discharge. . . I also had drastic mood swings and I would stay in bed all the day."* (S7)

*"I experienced continuous per vaginal bleeding for two consecutive months."* (S16)

*"After the first dose of vaccine, my menstrual cycle got delayed for two months, which made me very worried and depressed. . . I was so scared that I skipped taking my second dose of the vaccine and haven't taken it yet and I am not planning to take it either. In fact, I would suggest others not take it at all."* (S17)

*"Previously my cycles were very irregular and delayed. I used to have amenorrhea for 6 months. I even used to forget what periods feel like but after vaccination, it has improved. Now I have my periods after almost one and a half month."* (S10)

Four (23.5%) students, each, reported that these disturbances lasted for 3 months and between 3–6 months respectively, while only two (11.8%) students reported it to be for more than six months but less than a year. However, seven (41.2%) students reported these to be present for a year or more.

*"These changes have been there for the last one year but the severity of all these changes have reduced over time."* (S4)

**Theme 2: Reasons for menstrual disturbances; students' perspectives.** The majority (n = 9, 52.9%) of the students expressed their ignorance on what possibly would have caused these changes. Although all the others (n = 8, 47.1%) suggested that it might be a side effect of the vaccine, like side effects of other medication but five (29.4%) suggested hormonal changes after the COVID-19 vaccination as the main cause while one (5.9%) student connected it to an immune reaction. Two (11.8%) students, however, suggested that it is due to the stress, related to COVID-19 vaccination that has led to these problems.

*"The vaccine may cause hormonal changes which may, in turn, lead to such abnormalities."* (S2)

*"Although vaccines are safe it may be a side effect as every medicine has a side effect. Other than that, it may be stress related to the myths associated with the vaccine which may cause such abnormalities."* (S4)

*"I read somewhere on google that it could be some immune reaction."* (S7)

*"People also say that it can cause infertility. . . or it may be the stress and anxiety associated with the vaccine which can cause it."* (S17)

### Theme 3: Steps taken for the management

When asked about how they managed these issues, five (29.4%) of them reported that they decided to wait and see. Three (17.6%) of them consulted friends and family and were counseled, one (5.9%) used home-based (herbal) remedies and five (29.4%) consulted gynecologists for treatment. Two (11.8%), however, tried everything i.e., used home-based (herbal) remedies, consulted friends and family as well as consulted gynecologists for treatment. So, a total of 7 (41.2%) took specialist help to manage these issues.

*"I did not take any treatment as the symptoms resolved spontaneously."* (S3)

*"I initially took a homemade herbal medicine which my mother gave me but it did not help so then I went to a gynecologist who prescribed me oral progesterone for one month."* (S4)

*"I took painkillers and used a heating pad. Also, took a lot of rest during the first two days."* (S7)

*"I discussed all this with my friends and they told me that they were also going through the same changes so I thought it to be normal."* (S11)

*"I discussed it with my mother and she asked me to wait for three months. When it did not get better after three months, I tried some homemade herbal medicine and though I felt some improvement I went to a gynecologist who counseled me that these may be routine post-COVID-19 vaccination changes and advised lifestyle modification."* (S9)

### Theme 4: Post-COVID vaccination menstrual problems and mental health

The majority (n = 14, 82.4%) stated that these issues had an adverse impact on their mental health and they were anxious, stressed, worried, irritable, had disturbed sleep, nightmares, and headache and were depressed. Out of these 14, one visited a general physician and the other visited a psychiatrist for the mental health deterioration and they were counseled. Two (11.8%) felt no effect on their mental health and one (5.9%) due to improved menstrual cycle felt improved mental health status.

*"I became more irritable, angry and worried as I did not have such irregularities before the vaccination."* (S2)

*"It stressed me out and I had insomnia. It used to give me bad headaches throughout the day."* (S5)

*"No, instead I was happy to see that now my cycles got regular."* (S10)

*"After the vaccine, I can sense changes in my mood, 3 days ahead of the start of my periods. As I live in the hostel, I stayed more depressed, usually secluded myself and avoided talking to others. I was never like this before but ever since the disturbance in periods, I have been experiencing these. . . I have consulted a psychiatrist and he said these may be due to hormonal changes. All these were in the form of a chain of events after vaccination. I was actually worried because I read somewhere that this vaccine causes a problem with fertility."* (S7)

### Theme 5: Suggestions for the management of these problems

Students suggested more than one option for the management of such a condition. Almost half (n = 8, 47.1%) suggested consulting a gynecologist. Five (29.4%) of them were of the opinion that one has to stay updated about the side effects of the vaccine (medication) that is in use. Four (23.5%) suggested to use Self- Management and lifestyle changes and two (11.8%) were in the favor of stress management. Only one (5.9%) of the respondents thought that we should stop using the vaccine.

*"I would advise consulting a gynecologist as soon as anyone experiences such changes."* (S1)

*"If I had known what was coming, they maybe my response would have been much better. So, it is better to counsel everyone about the possible side effects, before the vaccination. Online search helped me and I educated myself and got to know what was happening to me."* (S7)

*"I think COVID vaccination should be stopped because it is causing major problems in the menstruation, or at least stop giving it till the cause of these irregularities is known because these changes affect mental health."* (S17)

*"I would suggest that do take supplements like iron and calcium so that even if one does experience such abnormalities, she should not be weak."* (S16)

*"Try to keep your self-stress free, as stress may be responsible for disturbing the menstrual cycles."* (S2)

The overall results from the quantitative and qualitative analysis revealed a significant impact of COVID-19 vaccination on menstrual irregularities and mental health which is depicted in Fig 1.

## Discussion

Recently there has been media hype on the effect of COVID-19 vaccination on the menstruation pattern. However, the UK's Medicine and Health care products Regulatory Agency (MHRA) states that the evaluation of yellow card surveillance does not support a link between changes in menstrual periods and COVID-19 vaccines [16]. The primary author, on the other hand, noticed an increase in the number of females presenting with menstrual irregularities after receiving the COVID-19 vaccination.

Our study reported that more than half of the students (53.7%) had menstrual cycle disturbances post COVID-19 vaccination, which is almost similar to the findings of Muhaidat et al., in the MENA region, who reported 66.3%, and Lee et al., in a web-based survey, reported 56.4% women presenting with menstrual irregularities after the vaccination [1, 8]. However, this is in contrast to the findings of Edelman et al. and Alvergne et al., that observed menstrual abnormalities after vaccination in 10% and 20% women, respectively [12, 17]. Although, the exact cause of these changes is not known, it is suggested that the vaccine induces immune-mediated thrombocytopenia which may be one of the causes [18]. Another suggested theory indicates that it can result in the loss of endometrial hemostasis explaining in heavy menstrual bleeding in these women [19].

In our study, the majority of the participants (66.4%) had no history of prior COVID-19 infection, which is similar to the study of Aolymet et al. who reported that 65.2% of their sample population also had no such history [20].

We found out that almost three fourth (74.6%) of females with menstrual disturbances had anxiety while only 12.5% of those had depression. Varying degrees of anxiety (60.1%, 33.7%,

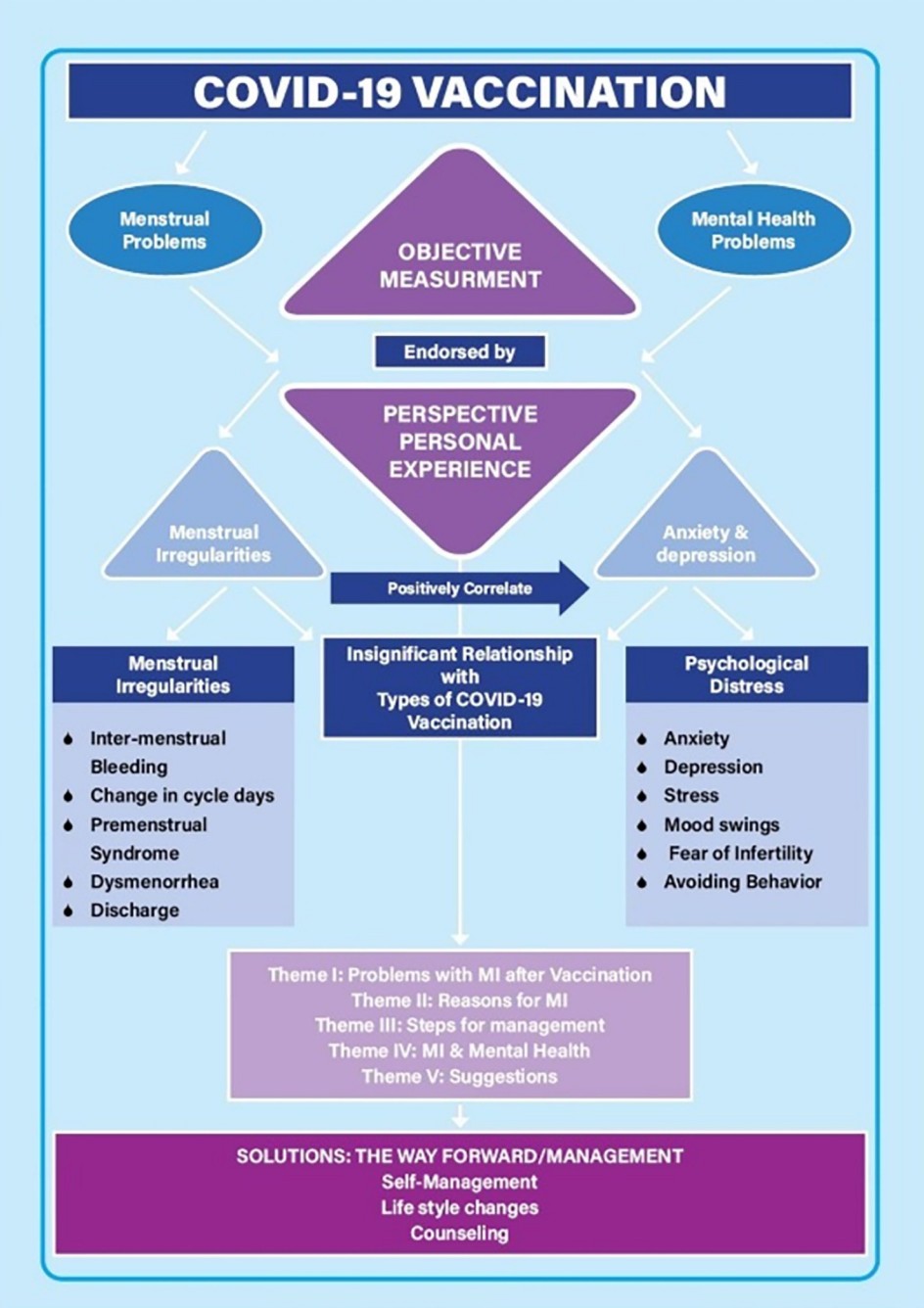

**Fig 1. Theoretical framework integrating the quantitative and qualitative results.**

13.76%, and 40.5%) and depression (65.5%, 31.5%, 13.5%, and 32.9%) have been reported in other studies showing that such abnormalities can worsen the psychological wellbeing of individuals and eventually disturb their quality of life [21–24].

The findings of our study concluded that vaccine type did influence the incidence of menstrual abnormalities but the vaccine-wise differences are in contrast to those reported by Lagana et al. who reported that 64.3% of women who received Moderna, 62.5% who received AstraZeneca, and 46.9% who received Pfizer reported alterations in their menstrual cycles [25].

In a study by Rizk et al., anxiety was reported in 23% of females receiving Sinopharm and in 46% receiving Pfizer vaccine, while in our study, it was reported in 21% receiving Sinopharm and in 5% receiving Pfizer vaccine [26]. In the same study, depression was reported in 22% of females receiving Sinopharm and in 54% receiving Pfizer vaccine, while in our study it was reported in 3.46% receiving Sinopharm and in 0.94% receiving Pfizer vaccine [26].

We then interviewed those students who suffered from menstrual abnormalities post COVID-19 vaccination and found that the majority of them suffered from disturbances of the frequency (58.8%) and flow (58.8%) followed by an increase in dysmenorrhea (41.2%). This is in contrast to the findings of a similar study done by Muhaidat et al. who reported that most of the women experienced changes in the regularity of menses (6.6%) followed by an increase in the frequency (4.5%) and worsening dysmenorrhea (4.3%) post COVID-19 vaccination and to a study done by Trogstad et al., where an increase in dysmenorrhea (16%) was the most common complaint, followed by heavy menstrual bleeding (15.3%) and prolonged bleeding days (14.3%) [1, 16].

In our study, 41.2% of the students reported that these changes lasted for more than a year and less than half the number of students (29.4%) decided not to use any treatment modality for the alleviation of these symptoms, the results are in contrast to another similar study which reported that these symptoms resolved within 2 months suggesting that these side effects are temporary and self-limiting and also reported that 65.5% of the participants did not resort to any modality for its treatment [1]. A qualitative study showed that there is social pressure to avoid discussing problems related to menstruation and that strongly influenced a woman's health-seeking behavior [27]. This pressure also explains the reasons why only a limited number of students (29.4%) in our study, went to an expert for an opinion while others decided to wait. Similarly, thinking of these changes as mild and self-limiting or lacking the awareness of risks associated with irregular menstruation, may have been the other reasons.

In the current study, the majority (52.9%) of the students did not know what caused these abnormalities. A similar study conducted on the women residing in the Middle East and North Africa region reported that the exact mechanism by which this vaccine causes menstrual changes are not known as yet and further research is needed [1]. It is suggested that the anxiety with the vaccine may cause these changes as stressors may activate the hypothalamic-pituitary-gonadal axis leading to a disruption of the regularity of the hormones and this is supported by a study, reporting that one or several menstrual issues are linked with a mental health condition such as anxiety and depression [28]. In our study, 5.9% of the students linked it to an immune reaction. This is similar to other studies that show that these symptoms are either a result of immunologic response or due to Inflammatory/immunologic reaction from adjuvants used in the vaccines [1, 6, 25].

In our study, 82.4% of the students suffered from mental health problems due to these menstrual abnormalities which are supported by a systematic review done on Asian women showing that menstrual issues are linked to numerous mental health issues such as fatigue, depression, and anxiety, and may lead to low quality of life [29]. Another study, in line with our findings, reported that menstrual problems are associated with psychological stress and women with high perceived stress were having high irregularity of menstruation and low quality of life (56.25%) [1].

## Limitations

Our sample, though large, may not be representative of the general female population of our region as this is taken from medical students studying in only one city. Also, as it is a self-reported survey, there may be chances of recall bias. However, the respondents were medical

students, well aware of their health-related issues, so the chances of recall bias would have been comparatively much less than the general population. Last but not the least, the lack of baseline data to allow for the comparison of menstrual cycle changes and mental health before and after the COVID-19 vaccination is a major limitation of the study.

## Conclusion

This study provides evidence that women receiving COVID-19 vaccines may experience menstrual cycle changes and such abnormalities may have an impact on the mental health of these women. Therefore, it is imperative to alert healthcare professionals about prior counseling of women about the possible side effects and should emphasize the need of seeking medical advice, if the symptoms are severe or persist for more than one cycle. Further research should be encouraged to investigate the short and long term effects of vaccination on reproductive health.

## Acknowledgments

We wish to thank the respondents for their willingness to provide information for this study. We also would like to thank Dr. Umema Zafar, Dr. Zahwa Salam, and Dr. Raheela Ameen for their efforts in collecting the data.

## Author Contributions

**Conceptualization:** Rabia Kareem, Mifrah Rauf Sethi.

**Data curation:** Rabia Kareem, Sumaira Inayat.

**Formal analysis:** Mifrah Rauf Sethi.

**Methodology:** Mifrah Rauf Sethi.

**Project administration:** Sumaira Inayat.

**Resources:** Muhammad Irfan.

**Supervision:** Muhammad Irfan.

**Writing – original draft:** Rabia Kareem, Mifrah Rauf Sethi, Sumaira Inayat.

**Writing – review & editing:** Muhammad Irfan.

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
