## [Decision Letter · Decision Letter 0]

26 Aug 2022

PONE-D-22-20959THE EFFECT OF COVID-19 VACCINATION ON THE MENSTRUAL PATTERN AND MENTAL HEALTH OF THE MEDICAL STUDENTS: A MIXED-METHODS STUDY FROM A LOW AND MIDDLE-INCOME COUNTRYPLOS ONE

Dear Dr. Irfan,

Thank you for submitting your manuscript to PLOS ONE. After careful consideration, we feel that it has merit but does not fully meet PLOS ONE’s publication criteria as it currently stands. Therefore, we invite you to submit a revised version of the manuscript that addresses the points raised during the review process.

I would like to emphasize Reviewer 1's recommendation on a stronger rationale for this study, such as the need to clarify the importance of the current study for this country. There are a number of mixed-method study types; please identify the type of study you are employing.

We look forward to receiving your revised manuscript.

Kind regards,

Ching Sin Siau

Academic Editor

PLOS ONE

Journal Requirements:

2. Please amend either the title on the online submission form (via Edit Submission) or the title in the manuscript so that they are identical.

Reviewers' comments:

Reviewer's Responses to Questions

**Comments to the Author**

1. Is the manuscript technically sound, and do the data support the conclusions?

Reviewer #1: Yes

Reviewer #2: Yes

2. Has the statistical analysis been performed appropriately and rigorously? 

Reviewer #1: Yes

Reviewer #2: Yes

3. Have the authors made all data underlying the findings in their manuscript fully available?

Reviewer #1: Yes

Reviewer #2: Yes

4. Is the manuscript presented in an intelligible fashion and written in standard English?

Reviewer #1: Yes

Reviewer #2: Yes

5. Review Comments to the Author

Reviewer #1: It is important to mention about what type of mixed method apply in this manuscript i.e sequential exploratory design, embedded, convergent etc. In addition, there are plenty of qoatation in the result section. Infact, it would be nice if you can interpete and provide the meaning. Furher, the discussion section does not intregate both result from qualitative and quantitative together.

Reviewer #2: COVID-19 vaccination is an important the impact of women`s menstrual patterns and subsequent mental health status., and its disturbances may negatively affect several aspects of women`s menstrual patterns and subsequent mental health status, etc. Therefore, the topic of the paper is important, and the questions raised by the study are worth an undertaking.

The study sample is large enough for appropriate statistical analysis and potential sound conclusions. The study design is a sequential-explanatory mixed-method study, which seem appropriate.

The tools selected by the authors seem relevant to the aims of the study. Although the findings from this study are interesting, there are several questions and problems that, in my opinion, should be resolved.

First, What is the number of full-dose vaccines in others country?

Second, it is not clear from the text, Cronbach’s alpha values of the scales used in the research should be added as well as the Cronbach's alpha values calculated in this study (Page 5 line 129 and 133).

Finally, mean mark should be revised using the symbols tab on the word page.

6. PLOS authors have the option to publish the peer review history of their article (what does this mean?). If published, this will include your full peer review and any attached files.

Reviewer #1: **Yes: **Yothin Sawangdee PhD

Reviewer #2: No

---

## [Decision Letter · Decision Letter 1]

19 Sep 2022

PONE-D-22-20959R1THE EFFECT OF COVID-19 VACCINATION ON THE MENSTRUAL PATTERN AND MENTAL HEALTH OF THE MEDICAL STUDENTS: A MIXED-METHODS STUDY FROM A LOW AND MIDDLE-INCOME COUNTRYPLOS ONE

Dear Dr. Irfan,

Thank you for submitting your manuscript to PLOS ONE. After careful consideration, we feel that it has merit but does not fully meet PLOS ONE’s publication criteria as it currently stands. Therefore, we invite you to submit a revised version of the manuscript that addresses the points raised during the review process.

I would like to encourage the authors to address Reviewer #1 and #2's comments in more detail. For example, I concur with Reviewer #1 that a further integration of quantitative and qualitative results is important, as this is a mixed-methods study, and not separate quantitative and qualitative studies. The authors responded that this has been addressed in the conclusions section. Usually, we do not add extra information in the conclusions section, merely summing up the important points which has been addressed in the results and discussion section. ==============================

We look forward to receiving your revised manuscript.

Kind regards,

Ching Sin Siau

Academic Editor

PLOS ONE

Journal Requirements:

Reviewers' comments:

Reviewer's Responses to Questions

**Comments to the Author**

1. If the authors have adequately addressed your comments raised in a previous round of review and you feel that this manuscript is now acceptable for publication, you may indicate that here to bypass the “Comments to the Author” section, enter your conflict of interest statement in the “Confidential to Editor” section, and submit your "Accept" recommendation.

Reviewer #1: All comments have been addressed

Reviewer #2: All comments have been addressed

2. Is the manuscript technically sound, and do the data support the conclusions?

Reviewer #1: Yes

Reviewer #2: Yes

3. Has the statistical analysis been performed appropriately and rigorously? 

Reviewer #1: Yes

Reviewer #2: Yes

4. Have the authors made all data underlying the findings in their manuscript fully available?

Reviewer #1: Yes

Reviewer #2: Yes

5. Is the manuscript presented in an intelligible fashion and written in standard English?

Reviewer #1: Yes

Reviewer #2: Yes

6. Review Comments to the Author

Reviewer #1: It would be helpful if this research team able to explain some more detail in their discussion when integrate both results between quantitative methodology and qualitative results together.

Reviewer #2: Dear Authors,

First of all, thank you for giving me the opportunity to review this manuscript.

COVID-19 vaccination is an important the impact of women`s menstrual patterns and subsequent mental health status., and its disturbances may negatively affect several aspects of women`s menstrual patterns and subsequent mental health status, etc. Therefore, the topic of the paper is important, and the questions raised by the study are worth an undertaking. But it is not clear from the text, Cronbach’s alpha values of the scales used in the research. It should be added as well as the Cronbach's alpha values calculated in this study.

7. PLOS authors have the option to publish the peer review history of their article (what does this mean?). If published, this will include your full peer review and any attached files.

Reviewer #1: **Yes: **Yothin Sawangdee

Reviewer #2: No

---

## [Author Response · Author response to Decision Letter 1]

15 Oct 2022

Integration of quantitative and qualitative results has been done in the form of a theoretical framework, and has been added and highlited in the manuscript.

---

## [Decision Letter · Decision Letter 2]

25 Oct 2022

THE EFFECT OF COVID-19 VACCINATION ON THE MENSTRUAL PATTERN AND MENTAL HEALTH OF THE MEDICAL STUDENTS: A MIXED-METHODS STUDY FROM A LOW AND MIDDLE-INCOME COUNTRY

PONE-D-22-20959R2

Dear Dr. Irfan,

We’re pleased to inform you that your manuscript has been judged scientifically suitable for publication and will be formally accepted for publication once it meets all outstanding technical requirements. 

Kind regards,

Ching Sin Siau

Academic Editor

PLOS ONE

Additional Editor Comments (optional):

Please refer to Reviewer #2's comments regarding some editorial issues which need to be corrected before it could proceed to the next stage of publication.

Reviewers' comments:

Reviewer's Responses to Questions

**Comments to the Author**

1. If the authors have adequately addressed your comments raised in a previous round of review and you feel that this manuscript is now acceptable for publication, you may indicate that here to bypass the “Comments to the Author” section, enter your conflict of interest statement in the “Confidential to Editor” section, and submit your "Accept" recommendation.

Reviewer #1: All comments have been addressed

Reviewer #2: All comments have been addressed

2. Is the manuscript technically sound, and do the data support the conclusions?

Reviewer #1: Yes

Reviewer #2: Yes

3. Has the statistical analysis been performed appropriately and rigorously? 

Reviewer #1: Yes

Reviewer #2: Yes

4. Have the authors made all data underlying the findings in their manuscript fully available?

Reviewer #1: Yes

Reviewer #2: Yes

5. Is the manuscript presented in an intelligible fashion and written in standard English?

Reviewer #1: Yes

Reviewer #2: Yes

6. Review Comments to the Author

Reviewer #1: Most of suggestion from peers' have been revising. The research methodology is appropriate now. This manuscript is publishable.

Reviewer #2: Dear Author

Minor revisions are recommended, as there is a typo in the reference section of the article please write ** p-value as a footnote under table 3.

Page 55 line 519; please revise the references according to the bibliography writing rule. (eg 2012;2: 345-9.)

Page 56 line 339: please revise the references according to the bibliography writing rule. (eg 2012;2: 345-9.)

Page 60 line 618: please write ** p value as footnote under table 3

7. PLOS authors have the option to publish the peer review history of their article (what does this mean?). If published, this will include your full peer review and any attached files.

Reviewer #1: **Yes: **Yothin Sawangdee

Reviewer #2: **Yes: **Gonca Buran
